# Novel Perspective of Hormesis in Evolution

**DOI:** 10.3390/biology15010012

**Published:** 2025-12-20

**Authors:** Marcela Vargas-Hernandez, Perla Valeria Munguia-Fragozo, Samantha de Jesus Rivero-Montejo, Diana Maria Amaya-Cruz, Juan Manuel Vera-Morales, Rosalia Virginia Ocampo-Velazquez, Israel Macias-Bobadilla, Irineo Torres-Pacheco

**Affiliations:** 1Faculty of Engineering, Campus Amealco, Autonomous University of Queretaro, Carretera Amealco Temazcaltzingo, km 1, Centro, Amealco de Bonfil 76850, Queretaro, Mexico; marcela.vargas@uaq.mx (M.V.-H.); valeria.munguia@uaq.mx (P.V.M.-F.); samantha.rivero@uaq.mx (S.d.J.R.-M.); diana.amaya@uaq.mx (D.M.A.-C.); juan.manuel.vera@uaq.mx (J.M.V.-M.); 2Laboratory of Biosystems Engineering, Faculty of Engineering, Campus Amazcala, Autonomous University of Queretaro, Carretera a Chichimequillas, km 1 S/N, El Marques 76265, Queretaro, Mexico; rosalia.ocampo@uaq.mx

**Keywords:** evolution, adaptation, adaptability, variation, variability, plasticity

## Abstract

It is necessary to contextualize hormesis within an evolutionary framework that incorporates the advances developed from Neo-Darwinism to contemporary evolutionary biology. The central problem addressed in this review is the lack of a conceptual integration between hormesis and evolutionary processes, which limits our understanding of its role in generating biological changes at different scales. Since hormesis actively contributes to the evolutionary process, we examine its relationship with key components such as adaptation, adaptability, variation, variability, and phenotypic plasticity. This analysis is essential for deepening our understanding of the genetic and phenotypic mechanisms that underline hormetic responses. Consequently, an updated definition of hormesis is proposed as a nonlinear adaptive process that depends not only on stimulus dose but also on exposure time, with particular emphasis on the cellular responses of organisms. By integrating hormesis into evolutionary thinking, this review offers a new perspective on how immediate stress responses may contribute to long-term evolutionary diversification and provides valuable insights for the design of future research strategies.

## 1. Origin of the Concept of Hormesis

Hormesis (from the Greek word hormaein, “to urge on” or “to excite”) was first described in the late nineteenth century [1]. The dose–response concept emerged from work conducted by Hugo Schulz, who observed that low concentrations of disinfectants stimulated metabolism in yeasts, in a dose–response behavior, while higher doses reduced it to the point of causing death [2,3]. In that same period, Schulz described that the growth of bacilli capable of inducing salmonellosis could be favored by the presence of veratrine at low doses, while higher doses were effectively used to treat the disease. From this, he concluded that veratrine benefited patients not by directly eliminating the bacteria, but because at low doses it induced an adaptive response in the individual capable of resisting infection [1,2]. He would soon come to call this phenomenon the Arndt–Schulz Law, which established that for every substance, small doses stimulate, moderate doses inhibit, and large doses kill [4,5]. It should be noted that the idea about this phenomenon was not completely new, as by the 16th century Paracelsus had stated that ‘All things are poison and nothing is without poison; only the dose makes a thing not a poison’, emphasizing that the toxicity of any substance depends on the administered dose and that even toxic substances could have beneficial effects in small doses. It was not until 1943 when Southam & Ehrlich [6] revisited the idea of the biphasic dose–response and gave it the term “Hormesis”, the name by which the phenomenon is known today. However, it was largely overlooked and by the late 1970s there were barely a few reports in the areas of pharmacology and toxicology. It was thanks to the work of Calabrese and other collaborators that the phenomenon regained relevance by identifying and documenting numerous studies on hormetic responses [7,8,9,10,11]. According to Calabrese & Baldwin [7], hormesis can be defined as “an adaptive response characterized by biphasic dose responses of generally similar quantitative features with respect to amplitude and range of the stimulatory response that are either directly induced or the result of compensatory biological processes following an initial disruption in homeostasis”. Different definitions of hormesis have been proposed over time (Table 1).

Reviewing the different definitions, this work proposes a new definition considering the exposure time and based on the responses at the molecular level: “Hormesis is a non-linear adaptive process through which cells, and consequently living organisms, respond to an increasing range of doses and time of exposure to a stressor. At low or moderate levels and/or short exposure times, the stimulus activates signaling and transduction networks that modulate gene expression, redox metabolism, and the synthesis of protective molecules, promoting biochemical and physiological adjustments and reaching a maximum value of the variable of interest associated with adaptation. Once a critical threshold of dose or exposure time is exceeded, defense mechanisms become saturated or dysregulated, and the beneficial response declines, leading to toxic or inhibitory effects”.

Currently hormesis studies are conducted in groups of individuals since doing so in a single individual is difficult because of measuring change and repeatable processes. This is done to consider homogeneity in genotype and epigenetic responses. Although individuals belong to the same species, there may be slightly different genotype changes or differences at the epigenetic level, which causes differential responses to the same stimulus. Figure 1 shows the hormetic curves at individual and population levels. The individual response represents the organism with the most efficient adaptation to the stress dose gradient, while the population curve reflects the average response of the population, integrating genetic variation and the action of natural selection. Due to this, the beneficial effect at the population level is lower than that present in the most adapted individual; furthermore, the population curve widens since there may be individuals who tolerate higher doses or conversely may present damage at lower doses.

On the other hand, because hormetic processes are increasingly recognized as evolutionarily conserved adaptive strategies, here we introduce a conceptual framework to understand how hormesis is linked to adaptation, adaptability, variation, variability, and phenotypic plasticity. Hormesis is not only a biphasic dose–response pattern, but a central mechanism through which organisms generate adaptive phenotypic diversity under changing environments.

## 2. Adaptation

Adaptation is a capacity resulting from a process that biological entities have to survive to a current condition or specific stress.

According to adaptation, Darwin argued that natural selection preserves heritable variants that enhance survival and reproduction, so organisms appear designed to maximize their Darwinian fitness [18,19]. In his view, pre-existing genetic variation among individuals allows the fittest to survive. In *On the Origin of Species*, Darwin raised several key points regarding adaptation; some of them are described below:The adaptation process is attributed to the cumulative action of natural selection, where slight modifications that favor individuals of a species tend to be conserved by adapting them to modified conditions.Physical adaptations are realized over time through natural selection, that is, through the survival of the fittest.Natural selection acts by adapting the present parts that vary in each being to their organic or inorganic life conditions, or by having adapted these during previous time periods, being aided by the increasing use or disuse of parts and under the influence of external life conditions and subject to different laws of growth and variation.Better-adapted individuals, assuming variability in a favorable sense, will tend to propagate in greater numbers than the less well-adapted ones.

From the above, it follows that adaptation confers on organisms a set of functional and evolutionary benefits that increase their probability of surviving, reproducing, and persisting in variable environments. These adaptive outcomes arise from the accumulation of small variations, with natural selection retaining those that provide functional advantages under selective pressures.

For his part, Ernst Mayr in *The Growth of Biological Thought* [20] noted that adaptation could no longer be considered a static condition, a product of a creative past, and became instead a continuing dynamic process. It should be clarified that this dynamic process of adaptation refers to the hormesis process. This is because stress stimuli at low doses (mild stress) elicit continuous adjustments of physiological, molecular, and phenotypic traits. In *The Strategy of the Genes: A Discussion of Some Aspects of Theoretical Biology*, Waddington [21] also considers adaptation as a process by which a structure or function comes into existence, and not merely as a static way of representing a structure or function under a particular set of conditions. Furthermore, he was the first to introduce the word canalization, the process that allows the organism temporarily to be independent of environmental fluctuations; however, at some point it must modify its development adaptively. Thus, in mixed populations, a stimulus can generate different modifications in individuals, and through selection, populations with their own characteristic responses can emerge. From this, it can be concluded that hormesis is a process that reflects the individual’s flexibility and adaptive capacity, while canalization acts by stabilizing phenotypes, buffering the effects of environmental perturbations to maintain constant development. If the hormetic response is very subtle, it may not appear as an evident change but rather as reinforced phenotypic stability, resembling canalization. In this sense, hormesis can operate within canalized developmental ranges that preserve phenotypic integrity while still allowing beneficial adjustments to mild stress. The hormetic response can shift the phenotype beyond canalized limits, partially breaking the canalization of the trait and releasing phenotypic variation upon which natural selection can act. Thus, while hormesis can modulate canalization and the range of phenotypic variability, it is also necessary to distinguish these effects from other forms of physiological regulation, namely homeostasis and homeorhesis. Homeorhesis, a concept proposed by Waddington [21], describes the property of a dynamic system to return to a particular trajectory after an external disturbance or despite the continuous presence of random noise. Hormesis can temporarily disrupt homeorhesis, but in some individuals, only canalization establishes mechanisms that can favor the individual’s adaptation. However, it is homeorhesis that corrects some alterations that may arise. Homeostasis, in turn, represents a dynamic equilibrium expressed in the normal ontogenetic condition of the organism and is related to time. Homeostasis is the result of the action of hormesis, canalization, and even homeorhesis; it is the dynamic balance of the individual’s ontogenetic state.

On the other hand, the Baldwin Effect [22] explained how the result of non-hereditary individual adaptation (behavioral or physiological) can, with time and natural selection, lead to heritable genetic changes. This process can be broken down into three steps: (1) Individual organisms interact with the environment to produce non-hereditary adaptations that allow them to survive. (2) In the population there are genetic factors that produce heritable characteristics similar to the individual modifications referred to in (1). (3) The genetic factors presented in (2) are favored by natural selection and tend to spread in the population through generations. However, there is still a lack of connection between points 1 and 2, that is, between non-hereditary individual adaptation and the emergence of genetic variants that mimic such adaptation. In relation to point (1), this interaction with the environment to generate adaptations constitutes in itself the hormetic process, which enables individual survival. The above considers that any adaptive change is necessarily a product of the environment and the genotype and will not necessarily appear in the next generation unless required. However, it is important to consider that the change itself is not heritable; what is inherited is the genetic potential to make it possible if needed in the next generation.

It is now understood that hormesis plays a crucial role in evolution, as it allows organisms to adapt to adverse conditions. However, evolution proper occurs when hereditary or epigenetic changes arise, primarily through mutations and the process of natural selection. If genetic adaptation is strictly defined—involving changes in DNA and selection—hormesis does not fit directly, as it acts on pre-existing genotypes, producing responses to low doses of stress that are not permanently or heritably encoded in DNA. However, Section 7 presents some evidence suggesting that hormesis can operate not only as a phenotypic mechanism but also through genetic mechanisms. Nevertheless, within the hormetic process, physiological or epigenetic adaptations can occur. Hormesis itself represents a form of stress-induced phenotypic plasticity; both processes act on the existing genotype without necessarily permanently altering the DNA. Furthermore, they can indirectly influence evolution by allowing organisms to survive while mutations arise, by enabling the genetic assimilation of the trait, or by shaping the landscapes of adaptation dependent on the environment.

Hormesis is essential for natural selection by favoring the most efficient adaptive responses, which increases the survival probability of individuals. An illustrative example is priming, where prior exposure to a stress stimulus prepares the individual to face subsequent adverse conditions with a more efficient adaptive response. As a result, that organism’s genotype has greater chances of remaining in the population, reproducing, and transmitting its genes to the next generation. This phenomenon can extend to future generations, contributing to the survival and persistence of the genetic lineage. Figure 2 illustrates a type of preconditioning or adaptive hormesis with cellular priming, where there are three consecutive exposures to stress, each followed by a recovery phase in which the response does not return to the initial level, but instead stabilizes at a higher point, demonstrating adaptive overcompensation. The curve reflects a priming process, in which each prior exposure better prepares the organism for the next challenge, creating a biological memory of stress. In summary, within Darwinian evolution, the hormetic process is a concept related to adaptation that occurs as a product of the combination of the individual’s genetic load and the changes that occur within the individual through environmental action. That is, hormesis represents the visible active component of adaptation, while the underlying genetic changes constitute the support that allows such adaptations to be heritable.

Darwinian adaptation is understood as a prescriptive process, that is, the result of genetic changes optimized by natural selection. In contrast, Magnon & Corbara [23] propose a proscriptive approach, in which organisms adapt within the limits of their current capacities without requiring genetic modifications. This perspective does not assume that adaptation achieves evolutionary perfection, but rather that it maintains functionality and viability in the face of environmental changes through phenotypic, physiological, behavioral, or cognitive adjustments, while also integrating processes of plasticity and systemic regulation. Hormesis constitutes a direct example of proscriptive adaptation, insofar as a low-dose stressor activates physiological, metabolic, or epigenetic pathways that are already available, operating upon a previously established genetic background within the individual.

## 3. Adaptability

Adaptability is the capacity of a species to adjust to environmental changes resulting from the potential conferred by its genetic variability in interaction with environmental and natural selection.

Mayr [20] linked it to the tendency of organisms to expand in number and mass, as well as to constantly invade and assimilate new environments, considering adaptability as the fundamental strategy to achieve this. Adaptability results from the potential conferred by genetic variability in interaction with environmental changes and natural selection. The above considers environmental changes as inducible and natural selection as the phenomenon of survival of the fittest in any environmental change. Adaptability is the product of both the species’ genetic pool and its mutation capacity, whose result allows organisms to have the capacity to adjust to different environmental changes or stressors, enabling them to survive in new environments. Hormesis strengthens individual adaptability by improving their capacity to face new challenges in nature, making them more resistant to future stress exposures. Therefore, in evolutionary terms, hormesis can be considered a factor that enhances adaptability by favoring the selection of individuals with more efficient responses, increasing phenotypic plasticity and therefore their capacity to face future challenges.

## 4. Variation

Variation refers to genetic and phenotypic differences between individuals of the same species in a specific environment or ambiental condition.

Darwin [18] emphasized several important points regarding variation, among which are the following:In the struggle for survival, individuals with unfavorable variations were eliminated while those with variations that made them more fit survived.The probability that a useful variation appears is directly related to the number of individuals; the greater the number of individuals, the greater the probability that a useful variation will appear.Variations, however slight, if they are to some degree profitable to the individual, will tend toward the preservation of individuals and will generally be inherited by offspring. For an important variation to exist, the population must be exposed to new environmental conditions for several generations. Sometimes similar variations originate under different environmental conditions and vice versa.

In relation to hormesis and the ideas proposed by Darwin, we can establish that the importance of variation lies in the fact that, on the one hand, it is the critical resource that enables natural selection to act, and, on the other hand, it increases the adaptive potential of species. The larger the number of individuals in a species, the greater its capacity to adapt. Individuals with greater variation generally possess a higher ability to survive in a specific environment; however, it is essential that within this broad variation there exist traits that confer a competitive advantage under particular environmental conditions. In the field of genetic improvement, the variation present in a population allows the selection of desirable traits even when these traits do not necessarily determine a competitive advantage. Hormesis, as a tool for adaptation, contributes to defining the range of feasible variation without requiring genetic changes. It also enhances the assessment of populations by revealing the range within which a genotype can normally survive under a specific environmental condition. Other authors have made various contributions to understanding the central characteristics of adaptation.

Although Gregor Mendel [24] did not extensively mention the concept of variability in his work, his contribution was fundamental to understanding biological variation by explaining in his laws of inheritance how genetic variation is transmitted and maintained between generations. This, in turn, allowed for the effective use of both phenotypic and genotypic variation in his experiments. This aspect is particularly relevant when analyzing hormetic curves in groups or populations of organisms.

Hugo de Vries, for his part [25], distinguished among mutations, defined as permanent changes with a direct impact on evolution; fluctuations, understood as minor and continuous variations lacking evolutionary relevance; and nutritional characters, referring to environmentally induced modified traits that are not heritable. In this sense, hormesis is more closely related to the latter, as it involves changes induced by the environment. On the other hand, Weismann, de Vries, and Dobzhansky [26] also maintained that changes in individuals caused by environmental variations are non-heritable. However, it is important to highlight that many of these environmental stress factors induce hormetic responses that operate through both genetic and phenotypic mechanisms.

Waddington [21,27] proposed that phenotypic changes induced by the environment during development can, over time, be fixed by natural selection, a process he called genetic assimilation. Under this perspective, it is possible that hormesis, by inducing useful adaptive responses to recurrent environmental stress, can contribute to the evolutionary fixation of initially plastic characteristics until transforming them into stable population traits. His ideas also marked the beginnings of epigenetics by proposing how genes and environment interact during development to shape the phenotype [28].

From the 1990s onward, modern epigenetics was consolidated, understood as the set of mechanisms through which the environment regulates gene expression without altering the DNA sequence. Among the mechanisms are DNA methylation, histone modification, and non-coding RNAs (ncRNA) [29,30]. Given that environmental factors can influence epigenetics, epigenetic changes expand the concept of biological variability. Like changes in DNA sequences, these modifications can shape the phenotype in a heritable manner, generating new variation patterns in individuals. It is assumed that epigenetics constitutes a key mechanism through which hormesis exerts its influence on genotypic variation and long-term evolution.

Although Darwin’s genetic understanding of adaptation held that adaptation is a gradual process, authors such as Bateson and de Vries argued that hereditary variations were discrete and of large effect, thus rejecting gradualism [31]. During the 1930s, Fisher [31,32] mathematically demonstrated that the combined action of numerous genes of small effect could generate continuous phenotypic variation, thereby reconciling Mendelian genetics with natural selection and establishing a quantitative framework for evolution. His infinitesimal model proposed that this variation arises from the additive contribution of multiple loci, although Fisher also recognized that adaptation does not always follow a universal trajectory toward a more favorable phenotype, as in the case of selfish genes. Subsequently, Fisher and Orr showed that, in addition to small-effect genes, there are mutations or genes of larger impact that can exert a disproportionate influence on adaptation, since they are more likely to produce significant phenotypic changes and accelerate evolutionary processes [31,32]. Since the 1980s, quantitative trait locus (QTL) analysis has made it possible to identify genomic regions associated with phenotypic differences between species or populations, demonstrating that adaptation often involves a small number of genes with moderate or large effects. Together with microbial experimental evolution, this has revealed that the first adaptive mutations tend to have the greatest effects on fitness, followed by additional mutations of smaller impact. Several efforts have been made to construct a mathematical theory of adaptation—including those of Wright, Kimura, Gillespie, Lande, and Orr himself—which seek to describe the distribution of mutational effect sizes, the rate of adaptive change, and the probability of fixation of beneficial mutations [31]. Other factors, such as epistasis, represent another key source of variability, acting as a modulator of evolutionary diversity by influencing the direction of species’ fitness and their adaptation [33]. Environmental factors that induce hormesis not only trigger adaptive responses but also modulate the expression of genes capable of interacting epistatically. In this way, hormetic conditions can reconfigure gene networks and alter interactions among loci, thereby modifying the effects of mutations on fitness. Moreover, these same factors can activate transposable elements, generating new mutations or genomic rearrangements that, in turn, become subject to epistatic regulation, thereby amplifying the genetic and phenotypic variability available for natural selection. It is also important to mention the contributions of Lynch [34], who argues that biological complexity is not always the result of adaptive natural selection but often arises from neutral or nearly neutral processes, such as mutation, genetic drift, and recombination. These processes generate the genetic variation upon which selection acts; however, in small populations, selection is less efficient and allows the random fixation of slightly deleterious mutations. He maintains that natural selection functions as a posterior filter rather than a creative force, and that complexity emerges mainly due to the relaxation of selective pressure rather than from a direct adaptive advantage. Overall, he proposes an evolutionary framework in which the combination of adaptive and neutral processes provides a better explanation for the origin and maintenance of biological complexity. The hormetic process can be aligned with Lynch’s ideas, since exposure to hormetic stressors may increase the rate of mutation or recombination, or modify epigenetic patterns, thereby enhancing the genetic and phenotypic variability available within a population.

## 5. Variability

Variability is the potential variation that a species may exhibit as a function of changing environments.

Lamarck had already expressed this idea by referring to the fact that the environment causes individuals to vary [35]. According to the ideas proposed by Darwin, the following can be established [18]:Great variability is evidently favorable since it increases the probabilities of the appearance of advantageous varieties and therefore natural selection can act more effectively.The change in living conditions is of utmost importance in producing variability; inheritance and reversion determine which variations will be lasting.Natural selection does not create variability; it only implies the conservation of varieties that appear and are beneficial to the environment.

According to the above, the usefulness of variability lies in the fact that it provides the possibility of selecting varieties that can adapt to different environments.

Mayr [20] establishes that variability arises from both genetic and non-genetic sources. There exist phenotypic changes, genetically determined, that are expressed favorably in specific environments and that are somatically induced by environmental conditions, without this implying a change in the genotype. In other words, one form of variability consists of gradual and reversible changes that arise in response to equally gradual environmental variations, reflecting a continuous phenotypic adjustment without any alteration of the genotype. It should be emphasized that such responses necessarily operate upon an existing genetic background.

Hormesis is part of and is a consequence of genetic variability that gives rise to physiological variability; both influence the organism and certain responses are favored or not as a consequence of environmental conditions. Hormesis management, through the type of stress, dose, and exposure, reveals the adaptive capacity that individuals possess, making visible the variability expressed in them. Individuals in a population may show greater or lesser hormetic capacity and as a whole explore a broader range of adaptive strategies, increasing their survival and evolutionary success under changing environmental conditions over time. Hormesis allows exposing existing variability, being the process for generating the raw material upon which natural selection acts. Figure 3 shows how phenotypic variability can manifest in different hormetic curves. Each curve corresponds to a different genotype indicating that different individuals or genetic variants can exhibit distinct hormetic responses to the same level of stress. It shows that both the adaptive capacity and the optimal stress range vary according to genetic variability.

## 6. Plasticity

Plasticity is the potential of an organism to change in order to survive in any environment, based on its genetic load, even before the challenge arises.

The concept of phenotypic plasticity emerges from the contributions of Johannsen [36]. Phenotypic plasticity is the idea of a reaction norm (Reaktionsnorm), a term coined by Woltereck [37]. He described this concept as the complete set of potentialities latent in a single genotype, evocable by the environmental circumstances of a developing organism. He also conducted the first study on adaptability in *Daphnia*, challenging the idea of saltationism and demonstrating that the same genotype could develop different phenotypes depending on environmental conditions. The above shows that the range of plasticity contributes to the adaptive capacity of the species. In this sense, hormesis can be considered a manifestation of the potential of phenotypic plasticity among individuals, showing how organisms are capable of adapting their phenotype to maximize survival under different environmental conditions. Figure 4 shows three examples representing the plasticity of different individuals in hormetic curves. In each graph, two genotypes are compared, demonstrating that the hormetic response is not uniform but depends on phenotypic plasticity and genetic variability.

Over time, numerous authors have enriched the understanding of the process of phenotypic plasticity. Schmalhausen [38] considered that plasticity is a mechanism that allows the organism to maintain stability. He introduced the concept of stabilizing selection, a process by which natural selection restricts the extremes of a phenotypic distribution, favoring intermediate values. Organisms must be shaped by stabilizing selection, conserving their most adequate capacity to respond to stress factors without exaggerating responses. The range of hormetic changes largely determines the degree of phenotypic plasticity expressed by organisms; natural selection acts on this variation by favoring individuals whose responses fall within the range of functional normality, while tending to eliminate extreme phenotypes that deviate from optimal limits.

On the other hand, Waddington [21] introduced the term “landscape passage” to illustrate how cells can follow different trajectories during their development and that these are influenced by environmental and genetic factors. Hormetic responses are part of this landscape passage process as they allow the organism to adjust its development and physiology in response to stress stimuli. Finally, he also highlighted the importance of plasticity: “Homeostatic maintenance of the ‘steady state’ of the organism in the face of changing environments is possible only thanks to a remarkable plasticity of the physiological machinery.” From this, we can conclude that plasticity allows both a diversity of hormetic responses but also an adequate homeostatic response when required.

Anthony David Bradshaw [39] distinguished physiological plasticity involving changes that can occur at any time, generally reversible and non-permanent, and morphological plasticity related to physiological changes that translate into lasting morphological effects. In the context of hormesis, this distinction is relevant: short-term with immediate responses (one or few exposures) involving purely physiological changes (e.g., enzymatic activity, hormonal modulation, antioxidant responses, etc.) and with repeated or prolonged exposures, morphological adjustments at the phenotypic level (changes in body proportions, cell number, tissue reinforcement, etc.). He notes that the plasticity of a character can arise from the lack of morphological (or physiological) homeostasis of that character, and at the same time, plasticity itself can lead to homeostasis in others. That is, if a character does not maintain its stability in the face of stress changes, it can be considered suboptimal, and plasticity allows it to vary to adapt. Hormesis facilitates that these less stable and unfavorable systems can adjust better, promoting beneficial adaptive adjustments.

According to Stearns [40], the relationship between evolution and reaction norm can be expressed in three forms of phenotypic plasticity: non-adaptive, where phenotypic change occurs but does not improve survival; maladaptive, where the plastic response decreases fitness; and adaptive, where the criterion for recognizing an adaptation is that the phenotypic change responds to a specific environmental signal and that there exists a clear functional relationship with that signal. In the hormetic curve, the beneficial response at low doses fits with what Stearns would call adaptive plasticity, whereas at high doses there is damage or a maladaptive response. At the beginning of the 2000s, Massimo Pigliucci [41], just like Stearns, revisited the concept of reaction norms, represented as an environment–phenotype graph, where the environmental parameter is placed on the abscissa and a measure of the phenotype on the ordinate. They can take more complex forms, for example polynomial functions [42,43]. In this sense, a hormetic dose–response curve constitutes an example of a nonlinear reaction norm (Figure 5). The graph compares three forms of reaction norms across different stress doses, showing how the adaptive response can vary depending on the biological model. Pigliucci [41] also points out that plasticity connects genotype, environment, and phenotype through epigenetic mechanisms, enabling adaptive adjustments that enhance the organism’s resilience—an effect to which hormetic phenomena contribute directly. He also explains the importance of an organism not canalizing its adaptive responses to a single environment but rather maintaining the capacity to survive and function in different types of environments, offering an evolutionary advantage by enhancing survival and resilience to stress.

Finally, Mary Jane West-Eberhard [44] establishes that plasticity is a trait subject to natural selection and evolutionary change, since both the direction and degree of plastic response are genetically variable. Within this framework, hormesis reinforces its role in evolution, illustrating how the environment acts simultaneously as a selective agent—where different phenotypes achieve varying degrees of survival and reproductive success—and as a developmental agent, significantly influencing the range of phenotypes produced by a given genotype. On the other hand, the evolution of adaptive plastic morphological responses requires that the signal operates early in ontogeny. A clear example is facultative seasonal polymorphisms involving morphology, which are governed by the duration of seasonal conditions, where both the intensity and duration of the environmental stimulus are decisive (e.g., photoperiod in insects). This implies that hormesis models should not only consider stressor doses but also the timing and duration of environmental signals. These aspects are particularly evident in species such as insects, whose life cycles are closely regulated by seasonal conditions. West-Eberhard highlights that an apparently small genetic change can trigger a complex set of co-expressed characters. In other words, a novel phenotypic trait can modify other structures in the individual, these being the product of plasticity and associated behaviors, amplified to accommodate and enhance a fundamentally simple but extreme phenotypic mutation. A mutation in a specific trait may induce a differential hormetic response in that same trait while simultaneously activating correlated hormetic responses in associated traits. This phenomenon produces a generalized hormetic effect, where multiple characters are adaptively coordinated and amplified. Thus, complex, coordinated, and adaptive phenotypes can arise rapidly from few genetic changes through correlated changes in the expression of plastic traits.

## 7. Molecular Mechanisms Involved in Hormetic Response

Hormesis fits within the paradigms of evolutionary theory, and to understand how such a response becomes manifested in functional terms, it is necessary to consider the molecular mechanisms underlying this process. It has been proposed that hormetic responses operate through both genetic and phenotypic mechanisms [45]. Several studies—both classical and recent—have demonstrated how environmental stress factors can increase the frequency of genetic recombination in organisms. This phenomenon is interpreted as an adaptive response that enhances genetic variability under unfavorable conditions. Another proposed mechanism is the induction of non-lethal mutations. Probably, at first, there is a hormesis reaction that is surpassed by the intensity of the factor that causes the mutation. Various stress factors, such as UV radiation, ionizing radiation, and exposure to heavy metals or oxidative chemical compounds, can generate reactive oxygen species (ROS), which in turn cause oxidative damage to DNA. This damage may induce point mutations or small genomic rearrangements that, although potentially harmful in excess, can be adaptive at low levels by activating DNA repair mechanisms and promoting the emergence of more resistant variants. At the epigenetic level, studies have shown that environmental stressors can induce chromatin modifications that are inherited at least temporarily through mitosis or even meiosis, conferring somatic or transgenerational memory of stress. Numerous studies also report that mild stress exposure activates different signaling pathways (Nrf2, HSF1, MTF-1, p53, FOXO) that induce the expression of genes encoding protective molecules (antioxidants, chaperones, metalloproteins, repair enzymes), thereby increasing cellular resistance to future, more intense stressors. Some of the studies supporting the mechanisms described above are summarized in Table 2.

Another important mechanism that plays a significant role in adaptation is transposable elements, or transposons, many of which respond to different types of environmental stress (Table 3). These elements can modify gene function and create genetic diversity, being the major source of variation [57]. There are evolutionary frameworks, such as Cognition-Based Evolution, in which the collective intelligence and cooperative behavior of cells operate through communication networks that utilize the previously described mechanisms, collectively coordinating adaptive responses to environmental stress factors. The systematic management of information is employed for the maintenance of preferred homeostatic boundaries among the varied participants, and these processes also enable genetic and cellular variations through self-referential natural informational engineering and cellular niche construction [58,59,60].

Other factors that may influence hormesis include the microbiome. This community of organisms affects a certain host species with beneficial, neutral, or detrimental effects. Given its importance, host-associated microbiomes have the potential to substantially contribute to the adaptation of the host–microbiome complex (the “metaorganism”). Although it is not yet fully understood how hosts and microbiomes jointly contribute to the adaptation of the metaorganism, some reports indicate that they can either improve or decrease fitness depending on the microbiome composition; therefore, adaptation to a novel environment can be jointly influenced by the host and the microbiome [67,68,69]. When the balance of the microbiota composition is altered by environmental stress (dysbiosis), a rapid adaptation occurs in the microbiome—faster than in the host—so the imbalance tends to diminish as both the host and the microbiome adapt to the new environment [70]. This disruption in the equilibrium between the host and the microbiome under stress conditions may act as a driver of adaptation. Moreover, some commensal bacteria of the microbiome can become harmful when environmental conditions change, contributing to the onset of diseases and, therefore, representing a stress factor for the host. The above is related to a hormetic process. Furthermore, even at the microbiome level, xenohormesis may occur—a phenomenon that refers to the transfer of defense responses from one species to another across trophic levels, including microorganisms (bacteria, nematodes, viruses, and fungi), insects, and plants [71].

Hormesis, being a generalizable process, is present in organs, cells, organisms, and at the population level; however, it originates from cellular responses. In this sense, mitohormesis is a biological response in which the induction of a reduced amount of mitochondrial stress leads to an increase in health and viability within a cell, tissue, or organism [72]. The activation of the mitohormetic response induces beneficial effects in different organisms, some examples of which are presented in Table 4.

## 8. Discussion

Unlike classical definitions, which focus solely on dose and the description of the biphasic effect, this proposal conceives hormesis as a nonlinear adaptive process, integrating not only the intensity of the stimulus but also emphasizing the duration of exposure. Furthermore, it provides a clear mechanistic basis by highlighting the activation of signaling networks, gene modulation, redox metabolism, and the synthesis of protective molecules, thereby linking hormetic responses to cellular physiology and homeorhesis. This perspective allows hormesis to be interpreted not merely as a dose–response pattern, but as an integrative and functional phenomenon that explains how organisms regulate their stability and adaptive capacity in response to environmental perturbations.

The present work attempts to provide a more comprehensive integration of the relationship between hormesis and evolution. In this context, the article aims to situate the concept of hormesis within the framework of conventional Darwinism and contemporary perspectives, highlighting its connection to fundamental evolutionary processes, including adaptation, adaptability, plasticity, variation, and variability. In addition, in order to understand how hormesis affects evolution, hormesis can be interpreted as a short-term adaptive mechanism, in which exposure to sublethal levels of stress activates responses at both the genetic and phenotypic levels. These mechanisms include the activation of transposons, the production of protective molecules, epigenetic processes, interactions with the microbiome, and the generation of mutations and their effects mediated by epistasis, all modulated by environmental factors. These responses not only allow organisms to maintain homeorhesis, but also generate beneficial adaptations, producing heritable changes that, in the long term, contribute to evolution. Evidence suggests that modifications in gene expression, epigenetic regulation, and cellular physiology can interact, amplifying the available genetic and phenotypic variability. In this way, hormesis acts as a direct link between immediate cellular mechanisms and the capacity of organisms to adapt, diversify, and evolve in response to changing environmental pressures.

## 9. Conclusions

The present review aimed to integrate the concept of hormesis within the framework of contemporary evolutionary perspectives. We discussed how hormesis, as a relevant source of evolutionary change, relates to key evolutionary components such as adaptation, adaptability, variation, variability, and phenotypic plasticity. In addition, we described the mechanisms through which hormesis may significantly contribute to evolutionary change. We believe that future studies should focus on developing experimental designs to investigate hormesis at both population and individual levels, incorporating factors such as dose, exposure time, genotype, plant phenological stage, and the concomitant use of elicitors, among others. These efforts should be accompanied by comprehensive analyses of responses to increasing stress levels at molecular and physiological scales, allowing for a more detailed elucidation of the underlying mechanisms.

## Figures and Tables

**Figure 1 biology-15-00012-f001:**
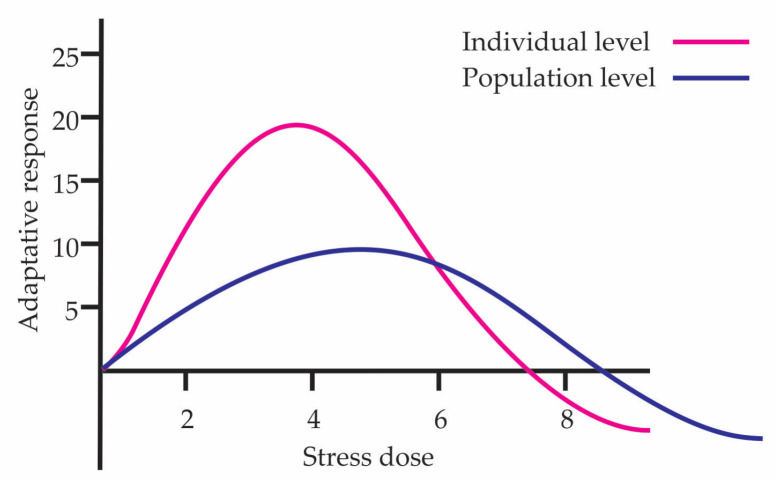
Hormetic curves at individual and population levels.

**Figure 2 biology-15-00012-f002:**
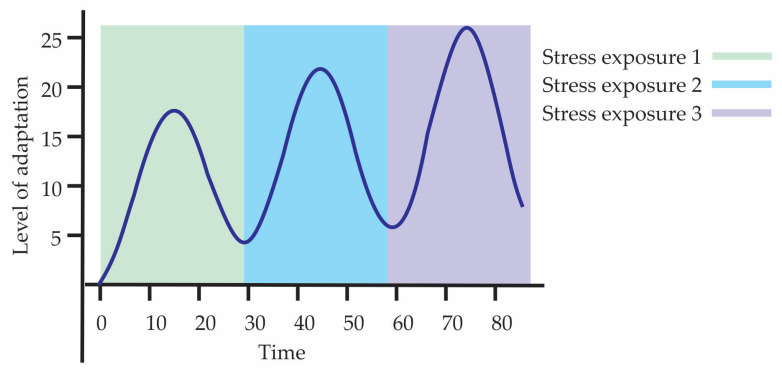
Hormesis and its temporal dynamics through adaptive priming.

**Figure 3 biology-15-00012-f003:**
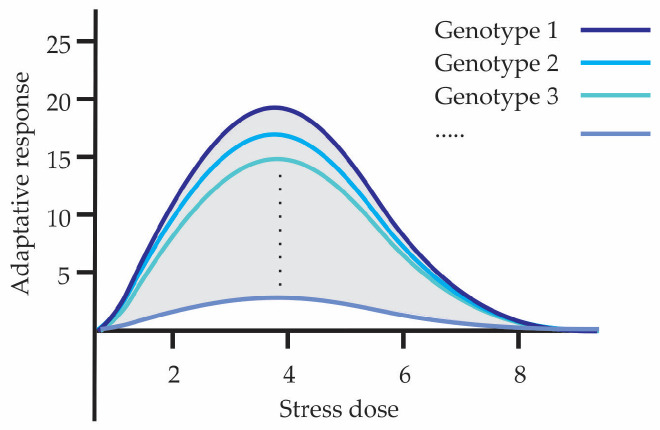
Ways in which phenotypic variability is visualized in hormetic curves.

**Figure 4 biology-15-00012-f004:**
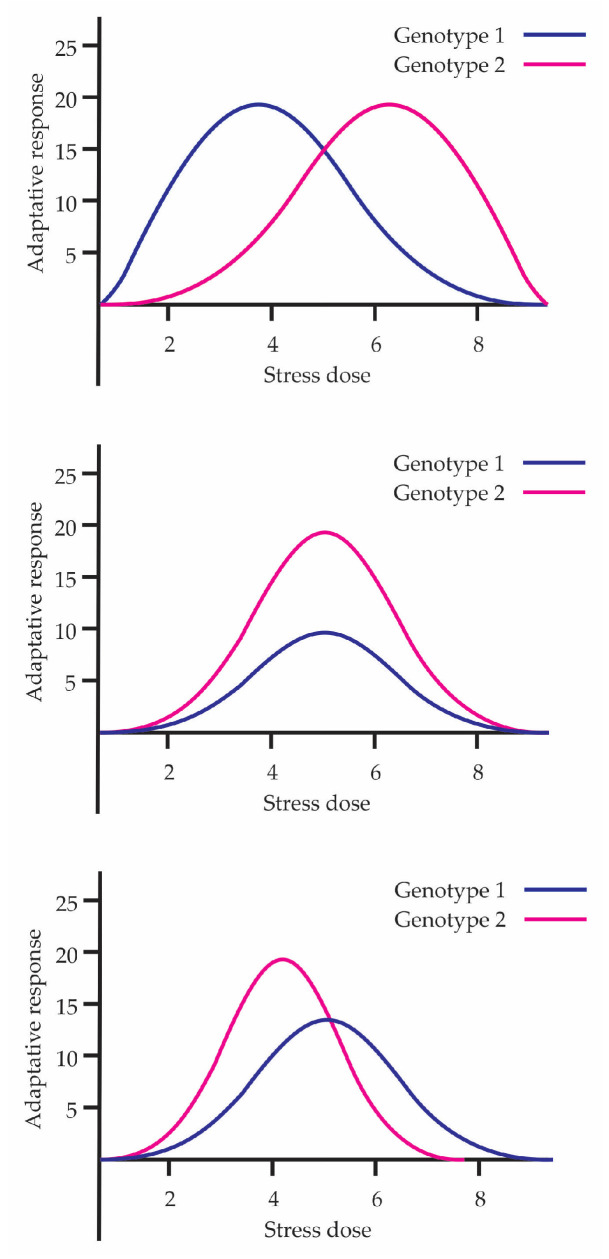
Representation of the plasticity of different individuals in hormetic curves.

**Figure 5 biology-15-00012-f005:**
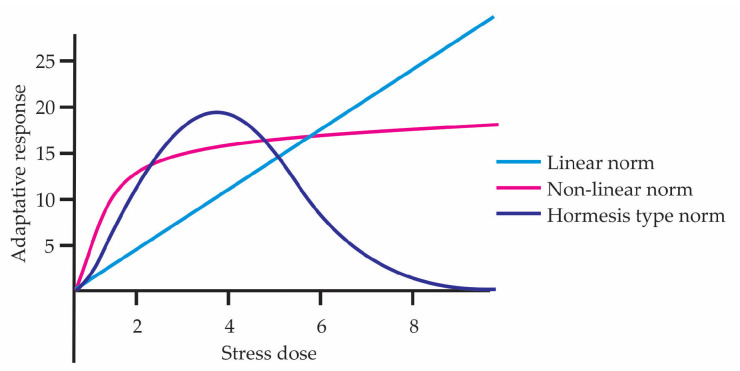
Reaction norm represented in linear, nonlinear, and hormetic forms.

**Table 1 biology-15-00012-t001:** Different definitions of hormesis according to various authors.

Author	Hormesis Definition
Paracelsus (16th Century) [12]	“What is there that is not poison? All things are poison and nothing is without poison. Solely the dose determines that a thing is not a poison.”
Arndt-Schulz’s law [2,3]	“For every substance, small doses stimulate, moderate doses inhibit, large doses kill.”
Southam & Ehrlich [6]	“A stimulatory effect of subinhibitory concentrations of any toxic substance on any organism.”
Stebbing [13]	Hormesis is the name given to the stimulatory effects caused by low levels of potentially toxic agents.
Christiani & Zhou [14]	Hormesis is a dose–response phenomenon characterized by either a U-shaped or an inverted U-shaped dose response depending on the different endpoints measured.
Calabrese & Baldwin [15]	Hormesis should be considered an adaptive response characterized by biphasic dose responses of generally similar quantitative features with respect to amplitude and range of the stimulatory response that are either directly induced or the result of compensatory biological processes following an initial disruption in homeostasis.
Mattson [16]	“A process in which exposure to a low dose of a chemical agent or environmental factor that is damaging at higher doses induces an adaptive beneficial effect on the cell or organism”.
Mattson & Calabrese [17]	Hormesis describes any process in which a cell, organism, or group of organisms exhibits a biphasic response to exposure to increasing amounts of a substance or condition (e.g., chemical, sensory stimulus, or metabolic stress); typically, low-dose exposures elicit a stimulatory or beneficial response, whereas high doses cause inhibition or toxicity.

**Table 2 biology-15-00012-t002:** Molecular and physiological mechanisms of hormetic stress responses.

Organism	Stress Factor	Response	Reference
*Arabidopsis thaliana*	Temperature, salinity, UV, osmotic, radiomimetic, oxidative.	Activation of somatic recombination and heterochromatic transcription.	[46]
*Saccharomyces cerevisiae*	Desiccation, nutrient starvation, temperature, overcrowding, depletion of resources, mechanical damage.	Activation of homologous recombination pathways.	[47]
*Streptococcus pneumoniae*	Antibiotics	Increase in genetic transformation (horizontal recombination).	[48]
*Escherichia coli*	Ionizing radiation	Adaptative mutations genes with function in the recovery of cells.	[49]
*Deinococcus radiodurans*	Ionizing radiation and desiccation	*recA*, *pprA*, *ddrA*, *ddrB* genes overexpressed or mutated, DNA repair.	[50]
*Arabidopsis thaliana*	Heat stress	Nucleosome loading and transcriptional silencing are restored upon recovery from heat stress but are delayed in mutants with impaired chromatin assembly functions.	[51]
*Mus musculus*	Olfactory stress, conditioning/Transgenerational	Transmission of behavioral and epigenetic response to F1 and F2 generations. Parental exposure to fear-associated odor altered methylation in olfactory receptor genes.	[52]
*Mammalian cells*	Oxidative stress (H_2_O_2_)	Induction of SOD, CAT, GPX/elimination of ROS; restoration of cellular redox balance.	[53]
*Arabidopsis thaliana*	Salt and drought stress	Salt stress is signaled via the SOS pathway where a calcium-responsive SOS3-SOS2 protein kinase complex controls the expression and activity of ion transporters such as SOS1. Osmotic stress activates several protein kinases including mitogen-activated kinases.	[54]
*Helix pomatia*	Cd and non-metallic stressors such as desiccation	Induction of Metallothionein Isoform/tolerance of the snail to heavy metals and other abiotic stressors.	[55]
*Triticum aestivum*	Water deficit	Proline accumulation/increased tolerance to water deficit.	[56]

**Table 3 biology-15-00012-t003:** Induction of transposable elements by stress factors that trigger beneficial adaptive responses.

Organism	Stress Factor	Transposon/Activation Mechanism	Response	Reference
*Arabidopsis thaliana*	Heat stress	Heat-activated retrotransposon ONSEN/generated a mutation in an abscisic acid (ABA) responsive gene.	Resulting in an ABA-insensitive phenotype in Arabidopsis, suggesting stress tolerance	[57]
*Zea mays*	Drought stress	TE families are significantly enriched for being located near genes with stress-responsive up-regulation and down-regulation.	Effect on up to 20% of the genes that are up-regulated in response to abiotic stress, and up to 33% of the genes that are expressed only in response to stress.	[61]
*Arabidopsis thaliana*	UV-B radiation	Diverse TE accumulation/transposition produces an exponential spread of TE copies, which rapidly leads to high mutation rates.	Increase the potential for rapid adaptation	[62]
*Arabidopsis thaliana*	Heat stress	*ONSEN*, an LTR-copia-type retrotransposon.	*ONSEN* exploits a conserved stress defense response.	[63]
*Schizosaccharomyces pombe* (*fission yeast*)	Heavy metals	LTR-retrotransposon (Tf1) insertions/stress conditions greatly increase TE mobility and insertions are targeted to promoters of stress-response genes	TE integration provided the major path to resistance.	[64]
*Drosophila melanogaster*	Xenobiotic exposure	FBti0019627 TE/insertion of TE FBti0019627 increases expression of gene CG11699 leading to elevated ALDH-III enzyme activity.	Xenobiotic stress Resistance	[65]
*Arabidopsis thaliana*	Heat stress	ONSET/heat-induced mobility of the heat-responsive retrotransposon ONSEN.	Increase phenotypic diversity and leads to drought-tolerant individuals in *A. thaliana.*	[66]

**Table 4 biology-15-00012-t004:** Mitohormetic responses mediated by stress factors.

Organism	Stress Factor	Mitochondrial Mechanism	Adaptative Response	Reference
*Carnorhabditis elegans*	Shengmai formula: Panax ginseng C.A.Mey. Radix (Pg)	Changes in CEP-1 expressions in a similar non-linear pattern in mev-1 and isp-1 mutants and cep-1 RNAi block the changes in mitochondrial dynamics.	Promoted the resistance to heat stress, oxidative stress, tended to increase the locomotion ability.	[73]
*Carnorhabditis elegans*	Green tea catechins	Hampered mitochondrial respiration complex I, isolated rodent mitochondria, which induced a transient drop in cellular ATP levels and temporary ROS burst.	Reduce fat content, enhanced ROS defense, and improved healthspan	[74]
Drosophila wild-type strains	KCN, antimycin, rotenone	Changes in the mitochondrial DNA compositions, increased of levels of mtROS, variation in complex I, and activations of AOX.	Increase in time of life span depending of the strain and sex	[75]
Drosophila larvae	Temperature stress	Response to mitochondrial complex I perturbation, redox-mediated stress signaling network impinging in part on the JNK pathway results in activation of the UPRmt.	Gene expression required in muscles with perturbed mitochondrial function.	[76]
*Mythimna separata*, *Drosophila melanogaster*	Acetyloxfenicince, tumor cells, heat stress.	Induces a transient depolarization of mitochondria, elevation of ROS and repression of lipid peroxidation.	Increased survival rates of test larvae after heat stress and nude mice significantly suppressed HCT cells.	[77]
*Arabidopsis thaliana*	Antimycin A	Trigger mitochondrial stress, activating retrograde signaling and inducing AOX1a gene and the activation of the NPR1 regulator as resistance induced by mitochondrial stress.	Trigger the appropriate response, and inducing the epigenetic memory of the stress to better react against future stressful conditions.	[78]
*Zea mays*	Saline stress	Alteration of NAD+ content by ORF355 expression, ORF355 inhibits mitochondrial ETC complex, function remains.	Superior growth potential and higher yield than those of the near-isogenic N-type line in saline fields.	[79]
*Nothobranchius furzeri*	Calorie/dietary restriction	ROS levels trigger mitohormesis by upregulating the sirtuin signaling pathway.	Improvement in muscle health.	[80]
*Rhodosporidium toruloides*	Glucose nutrimental stress.	Varying longevity mechanisms exhibited by different geroprotectors in cells by the up- and down-production of ROS.	Apoptosis and increased lifespan.	[81]

## Data Availability

No new data were created or analyzed in this study. Data sharing is not applicable to this article.

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
