# Peer review of "Novel Perspective of Hormesis in Evolution"

_biology, 2025, doi:10.3390/biology15010012_

Round 1

Reviewer 1 Report

Comments and Suggestions for Authors

Comments IN THIS current paper, the HORMESIS process is recognized as a key mechanism underlying Darwin and Wallace's theory of evolution, and to explore its relationship with other processes linked to natural selection. Advances in knowledge of various metabolic processes associated with evolution give rise to the emergence of new concepts and the exploration of their interrelationships. Besides hormesis, fundamental notions of processes or characteristics such as adaptation, adaptability, plasticity, variation, and variability, among the main ones, are explained. the ideas originated by Darwin are reviewed in evolutionary biology, while several concepts have been discussed, and others have emerged in relation to the phenomena described above. The paper tends to establish consistent, updated, and expanded definitions that allow the integration of hormesis with evolutionary processes The contribution is outstanding for  explanation of the process, though is very general, shows some potential . Collectively, the above comments are good, and neat, figures including Hormetic curves at individual and population levels,  Hormesis and its temporal dynamics through adaptive priming, while addressing Darwin's theories, several key points regarding adaptation show the analysis in good manner, while natural selection can favor genotypes that respond positively to stress factors. The conclusions are very consistent with the evidence and some can conclude that the topic is original and relevant to the field, environmental changes can induce adaptive responses and through time and natural selection, can be inherited by the population to survive environmental changes that do not revert . only Please check carefully English language.

Comments on the Quality of English Language

some improvement needed

Author Response

Comments IN THIS current paper, the HORMESIS process is recognized as a key mechanism underlying Darwin and Wallace's theory of evolution, and to explore its relationship with other processes linked to natural selection. Advances in knowledge of various metabolic processes associated with evolution give rise to the emergence of new concepts and the exploration of their interrelationships. Besides hormesis, fundamental notions of processes or characteristics such as adaptation, adaptability, plasticity, variation, and variability, among the main ones, are explained. the ideas originated by Darwin are reviewed in evolutionary biology, while several concepts have been discussed, and others have emerged in relation to the phenomena described above. The paper tends to establish consistent, updated, and expanded definitions that allow the integration of hormesis with evolutionary processes The contribution is outstanding for  explanation of the process, though is very general, shows some potential . Collectively, the above comments are good, and neat, figures including Hormetic curves at individual and population levels,  Hormesis and its temporal dynamics through adaptive priming, while addressing Darwin's theories, several key points regarding adaptation show the analysis in good manner, while natural selection can favor genotypes that respond positively to stress factors. The conclusions are very consistent with the evidence and some can conclude that the topic is original and relevant to the field, environmental changes can induce adaptive responses and through time and natural selection, can be inherited by the population to survive environmental changes that do not revert . only Please check carefully English language.

Response 1:

We appreciate the reviewer's comments; the manuscript was then submitted for English review by the same Publisher MDPI.

Reviewer 2 Report

Comments and Suggestions for Authors

The article is a well-written opinion piece and a general review of the historical sequence of evolutionary ideas relating to the applicability of hormesis to evolutionary processes. Hormesis is appealing because it enhances an organism's adaptive flexibility through conditional low-level exposures to hazards. In such a framework, those organisms that survive that exposure are being selected since they are the ones that have the 'correct' genes that encode for environmentally appropriate heat-shock proteins, antioxidant enzymes, neurotransmitters, hormones, etc., or that mediate behavioral responses to environmental stressors (neurotransmitters, hormones, muscle cell growth factors, etc).

The major limitation is that the manuscript is highly conventional and covers very little new ground. Furthermore, there are some significant issues that relate to clarity, conceptualization, and putative novelty.

First, with respect to conception, the manuscript's premises and discussion are very 'old school', firmly ensconced in 20th-century evolutionary biology. The reviewer acknowledges that the viewpoints expressed in the article are likely to sit quite comfortably with most evolutionary biologists, many of whom think little about cellular processes. However, cellular biologists will find it demonstrably lacking since it does not recognize that hormesis is necessarily a cellular process.

Consequently, any modern discussion of hormesis must include a discussion of the relevance of cell-signaling pathways, cellular enzymes, transcription factors, and cytoprotective proteins, or how cellular processes such as cell proliferation, growth patterns, tissue repair, or cell-based adaptive behaviors might contribute.

Such a discussion need not be in any great depth, but if there is to be any putative 'maximal adaptive peak', it must travel through cellular mechanisms, just as any epigenetic change must necessarily occur at the level of individual cells.

Secondly, with respect to clarity, the authors offer a new definition of hormesis: "The process of biochemical and physiological changes generated in an organism within a spectrum of increasing doses of a stress factor in the environment until reaching a point where the maximum value of the most representative variable of possible adaptation or the maximum value of a variable of interest is expressed and then declines.”

Frankly, this is not only distinctly unwieldy but nearly impenetrable. The terms are unclear, making it difficult to envision their application. It is not an improvement over many prior definitions. This definition needs to be reformulated so that its meaning is transparently clear. Furthermore, it is also unclear how this definition, as the reviewer is able to interpret it, is different from the J-shaped or U-shaped curves that are conventionally used to define hormesis. Interestingly, despite being noted in the conclusion, the time element is absent from the new definition.

The reviewer encourages the authors to explicitly define the differences between their prior concepts of homesis and their novel perspectives. This might be best achieved by explaining how such a new definitional standard can be applied to biology, allowing for direct understanding. As it stands, the old definitions might have been insufficient, but they were clear. Certainly, these talented authors can match prior frameworks. Specifically, what is meant by 'maximum adaptive value' and how is that determined? What makes one thing a 'variable of interest' and not another? It is unclear to the reviewer how applying these concepts significantly changes our understanding of hormesis. Moreover, the reviewer does not see how "the present work offers a more complete integration," nor does the mention of a 'priming' role in evolution by hormesis constitute any level of novelty, since this is well-acknowledged in the literature.

Thirdly, in considering novelty, the authors assert that the paper distinguishes itself from previous papers on hormesis. For example, the authors propose (seeming to imply that this is novel)  that "species survival in evolutionary biology is understood as the result of the joint action of three essential factors": the existence of variation, the existence of natural selection, and time. However, this proposition is exactly what is conventionally believed. The reviewer does not see how this is novel. Certainly, the existence of variation is uncontroversial. The pertinent issue is the origin of variations. Undoubtedly, environmental strictures expunge organisms that cannot cope with conditions,which has been termed natural selection but is merely environmental filtering. Further, the concept of evolution is necessarily a time-dependent process, which is why it's called 'evolution'. Moreover, the time-dependency of hormesis is broadly and consistently recognized in the literature. Where is this novel?

The reviewer has struggled to understand where novelty lies since the entire process is presented in the terms of conventional NeoDarwinism. This criticism also relates to the highly conventional attitudes to the sources of variation, how these variations spread through populations, and the role of natural selection.

Two issues arise from this concern. The first is the general attitude throughout the article that variation is fundamentally a genetic issue. Of course, genetic variations are important sources of variation, but by no means the only ones. Furthermore, in a modern review, epigenetic mechanisms deserve more than a passing mention.

The crux of any evolutionary debate is the issue of how variations arise in individuals and populations. This debate has recently focused on the growing consensus that the prior NeoDarwinian assumption — that variations arise through random genetic variations, mostly due to replication errors —has been decisively eclipsed. Most notably, it is becoming increasingly recognized that natural selection is not a selective process per se. It is a filter of precedent variations. Variations arise, and they are either consonant with environmental conditions or they are expunged. So, by the defining terms of natural selection, it is a post-facto filter and not a selective agency. It is not clear from the manuscript whether the authors actually grasp this crucial factor. Any creativity in evolution can only spring from the source of variation, not the filtering of its outputs, no matter what has been previously believed.

The authors should articulate the varied sources of hormetic variation to include the great variety of epigenetic sources and the proteome (e.g. proteomic constraint, translation effects, network interactions, domain shuffling, etc.). None of this needs to be lengthy.

For example, the authors might want to become acquainted with the peer-reviewed concepts of natural genetic editing as sources of variation. Furthermore, the authors might be interested in learning about a peer-reviewed alternative to NeoDarwinism (Cognition-Based Evolution ) that proposes that adaptation proceeds through the collective action of intelligent cells. In this framework, intelligent cells solve problems (environmental stresses) by deploying their cellular tools as adaptive responses. In this way, hormesis can be seen as problem-solving and adaptive at one level and counterproductive at higher thresholds, such as we commonly experience with exercise or many foods or beverages. This is mentioned so that the authors can see that evolution is moving beyond traditional NeoDarwinism, and they might want to become familiar with this new direction. There is no review requirement to include this alternative perspective in an amended article.

With respect to variation, and considering that this is a modern review intent on separating itself from previous ones, a further deficiency is the absence of any discussion of the role of the microbiome. The microbiome has a well-acknowledged role in hormesis, relating to metabolic products and signaling molecules, mitohormesis, and distinct contributions to epigenetic modifications. In particular, the paper would benefit from discussing the concept of pathobionts and their role in adaptation. Indeed, the entire concept of dysbiosis can be conceptually linked to hormetic response systems in context.

A few minor criticisms where amendments should or might be considered are noted to follow:

a) The reviewer is unclear why Weismann's ideas are being discussed, since they have been proven wrong, see

Noble, D. (2021). The illusions of the modern synthesis. Biosemiotics, 14(1), 5-24. If these are going to be discussed, then it is pertinent to mention that his notions are now considered invalid.

b) The authors state: "It also explicitly includes that the stress factor is environmental, which broadens the conceptual framework beyond chemical or pharmacological agents." Aren't chemical and pharmacological agents in the environment? Aren't some of them actual or potential mutagens? The reviewer does not understand the logic of this statement. Furthermore, the integration of hormesis into an evolutionary framework, extending it beyond toxicology, has been going on for decades, as acknowledged by the authors in their historical review.  How is this novel?

c) Just as a point of discussion, the authors mention Darwin's attitude about selection and 'perfection'. They attribute this to Darwin: " The final result of natural selection tends to become more and more perfected in relation to environmental conditions." It is uncertain whether the authors are aware that this flat statement misleads the reader unless further context is provided. Darwin meant this term as a relative one, … one organism relative to another, but it was not intended to suggest actual perfection.

In fact, the notion that evolution leads to perfection is clearly misguided. If organisms are 'perfectly fit' within any given set of environmental conditions, they necessarily have sacrificed robustness. Just to note, evolution is never about perfection, nor is it actually 'survival of the fittest', a trope that has long outlived its usefulness. Evolution is always about being 'fit enough to survive, ' which better maintains the potential variations that remain 'on-call' to meet unpredictable shifts in environmental conditions.

d) p. 11 " The importance of changes within the environment in which individuals develop for the production of phenotypic variability is imminent." Should that be 'eminent'. How is it imminent?

e) Also, in the figures, shouldn't 'genotipe' be 'genotype'?

In sum, an interesting article, which at its base is a useful review of historical thinking about hormesis. To be suitable for publication with respect to novelty, the authors need to consider several amendments, relating to improving the clarity of their conceptualization and creating a more workable definition. One suggestion for that pathway would be to focus on cellular mechanisms. The authors might find others. As an example, in the discussion and conclusion, the authors indicate that their approach differs from others by emphasizing a definition that "delves deeper into the sequence of internal changes the organism, emphasizing the temporal and progressive nature of the response." At issue is that the manuscript, as it is written, does not demonstrate how those processes operate within a 'new' definition, and further, that definition remains problematic.  For example, this statement is made: " A local stimulus can induce beneficial changes at the systemic level, showing that hormesis is a phenomenon that transcends the local response from which it originates." Certainly, but how does this happen within this new definitional context, and how is that different, since this is the essence of plasticity?  In that way, the novelty of any new framework would be more clearly revealed to the readership, and the definitional terms would crystallize. In the reviewer's opinion, the best approach would be to include additional information that focuses on cellular mechanisms that better illustrate how hormesis relates to adaptive flexibility.

Author Response

Comment 1

The article is a well-written opinion piece and a general review of the historical sequence of evolutionary ideas relating to the applicability of hormesis to evolutionary processes. Hormesis is appealing because it enhances an organism's adaptive flexibility through conditional low-level exposures to hazards. In such a framework, those organisms that survive that exposure are being selected since they are the ones that have the 'correct' genes that encode for environmentally appropriate heat-shock proteins, antioxidant enzymes, neurotransmitters, hormones, etc., or that mediate behavioral responses to environmental stressors (neurotransmitters, hormones, muscle cell growth factors, etc).

The major limitation is that the manuscript is highly conventional and covers very little new ground. Furthermore, there are some significant issues that relate to clarity, conceptualization, and putative novelty.

First, with respect to conception, the manuscript's premises and discussion are very 'old school', firmly ensconced in 20th-century evolutionary biology. The reviewer acknowledges that the viewpoints expressed in the article are likely to sit quite comfortably with most evolutionary biologists, many of whom think little about cellular processes. However, cellular biologists will find it demonstrably lacking since it does not recognize that hormesis is necessarily a cellular process. Consequently, any modern discussion of hormesis must include a discussion of the relevance of cell-signaling pathways, cellular enzymes, transcription factors, and cytoprotective proteins, or how cellular processes such as cell proliferation, growth patterns, tissue repair, or cell-based adaptive behaviors might contribute. Such a discussion need not be in any great depth, but if there is to be any putative 'maximal adaptive peak', it must travel through cellular mechanisms, just as any epigenetic change must necessarily occur at the level of individual cells.

Response 1:

The authors acknowledge that hormesis, although present in organisms, originates from molecular responses within cells. Therefore, a section (section 7) describes the molecular mechanisms involved in the hormetic response, considering both genetic and phenotypic aspects. A table is included with various examples of how stress factors impact cellular levels.

Comment 2

 Secondly, with respect to clarity, the authors offer a new definition of hormesis: "The process of biochemical and physiological changes generated in an organism within a spectrum of increasing doses of a stress factor in the environment until reaching a point where the maximum value of the most representative variable of possible adaptation or the maximum value of a variable of interest is expressed and then declines.”

Frankly, this is not only distinctly unwieldy but nearly impenetrable. The terms are unclear, making it difficult to envision their application. It is not an improvement over many prior definitions. This definition needs to be reformulated so that its meaning is transparently clear. Furthermore, it is also unclear how this definition, as the reviewer is able to interpret it, is different from the J-shaped or U-shaped curves that are conventionally used to define hormesis. Interestingly, despite being noted in the conclusion, the time element is absent from the new definition.

The reviewer encourages the authors to explicitly define the differences between their prior concepts of homesis and their novel perspectives. This might be best achieved by explaining how such a new definitional standard can be applied to biology, allowing for direct understanding. As it stands, the old definitions might have been insufficient, but they were clear. Certainly, these talented authors can match prior frameworks. Specifically, what is meant by 'maximum adaptive value' and how is that determined? What makes one thing a 'variable of interest' and not another? It is unclear to the reviewer how applying these concepts significantly changes our understanding of hormesis. Moreover, the reviewer does not see how "the present work offers a more complete integration," nor does the mention of a 'priming' role in evolution by hormesis constitute any level of novelty, since this is well-acknowledged in the literature.

Response 2:

We appreciate your valuable comments. The definition of hormesis has been revised to include the underlying cellular mechanisms and the temporal dimension, without going into excessive detail to avoid making the definition unnecessarily complex. The latter aspect is particularly important, since current models typically consider only the dose of the stress factor and the corresponding survival response. These modifications and clarifications are explained in the Discussion section, This section describes the main characteristics of our definition.

In addition, there is evidence of response curves that are not strictly biphasic; however, to avoid controversy, this issue was not discussed further. For these reasons, we believe that a definition based on cellular responses leading to the improvement of an adaptive variable is more appropriate. Moreover, if parameters such as dose, response, and exposure time are considered in the model, we suggest that the shape of the curve might be modified accordingly.

The term “maximum adaptive value” explicitly refers to the value reached by the variable (or variables) associated with adaptation in response to a given level of stress, as now indicated in the revised definition.

It is also important to note that, in the context of hormesis, the variable selected by the researcher or agronomist does not always reflect a better physiological or health condition of the plant. For example, a lack of light may induce excessive stem elongation, resulting in greater plant height. However, such growth does not necessarily indicate a healthier condition; on the contrary, in many cases, plants with shorter stems exhibit a more robust structure, balanced development, and consequently higher survival. Similarly, a hormetic response that increases a specific variable (such as height or biomass production) does not automatically imply an overall improvement in health or adaptability of the organism, but rather a specific adaptive response to the stress factor. Nevertheless, we believe that this topic still requires further analysis and was not addressed in the present work, as it may be better explored in future reviews.

Comment 3:

Thirdly, in considering novelty, the authors assert that the paper distinguishes itself from previous papers on hormesis. For example, the authors propose (seeming to imply that this is novel)  that "species survival in evolutionary biology is understood as the result of the joint action of three essential factors": the existence of variation, the existence of natural selection, and time. However, this proposition is exactly what is conventionally believed. The reviewer does not see how this is novel. Certainly, the existence of variation is uncontroversial. The pertinent issue is the origin of variations. Undoubtedly, environmental strictures expunge organisms that cannot cope with conditions,which has been termed natural selection but is merely environmental filtering. Further, the concept of evolution is necessarily a time-dependent process, which is why it's called 'evolution'. Moreover, the time-dependency of hormesis is broadly and consistently recognized in the literature. Where is this novel?

The reviewer has struggled to understand where novelty lies since the entire process is presented in the terms of conventional NeoDarwinism. This criticism also relates to the highly conventional attitudes to the sources of variation, how these variations spread through populations, and the role of natural selection.

Two issues arise from this concern. The first is the general attitude throughout the article that variation is fundamentally a genetic issue. Of course, genetic variations are important sources of variation, but by no means the only ones. Furthermore, in a modern review, epigenetic mechanisms deserve more than a passing mention.

The crux of any evolutionary debate is the issue of how variations arise in individuals and populations. This debate has recently focused on the growing consensus that the prior NeoDarwinian assumption — that variations arise through random genetic variations, mostly due to replication errors —has been decisively eclipsed. Most notably, it is becoming increasingly recognized that natural selection is not a selective process per se. It is a filter of precedent variations. Variations arise, and they are either consonant with environmental conditions or they are expunged. So, by the defining terms of natural selection, it is a post-facto filter and not a selective agency. It is not clear from the manuscript whether the authors actually grasp this crucial factor. Any creativity in evolution can only spring from the source of variation, not the filtering of its outputs, no matter what has been previously believed.

The authors should articulate the varied sources of hormetic variation to include the great variety of epigenetic sources and the proteome (e.g. proteomic constraint, translation effects, network interactions, domain shuffling, etc.). None of this needs to be lengthy.

For example, the authors might want to become acquainted with the peer-reviewed concepts of natural genetic editing as sources of variation. Furthermore, the authors might be interested in learning about a peer-reviewed alternative to NeoDarwinism (Cognition-Based Evolution ) that proposes that adaptation proceeds through the collective action of intelligent cells. In this framework, intelligent cells solve problems (environmental stresses) by deploying their cellular tools as adaptive responses. In this way, hormesis can be seen as problem-solving and adaptive at one level and counterproductive at higher thresholds, such as we commonly experience with exercise or many foods or beverages. This is mentioned so that the authors can see that evolution is moving beyond traditional NeoDarwinism, and they might want to become familiar with this new direction. There is no review requirement to include this alternative perspective in an amended article.

With respect to variation, and considering that this is a modern review intent on separating itself from previous ones, a further deficiency is the absence of any discussion of the role of the microbiome. The microbiome has a well-acknowledged role in hormesis, relating to metabolic products and signaling molecules, mitohormesis, and distinct contributions to epigenetic modifications. In particular, the paper would benefit from discussing the concept of pathobionts and their role in adaptation. Indeed, the entire concept of dysbiosis can be conceptually linked to hormetic response systems in context.

Response 3:

The authors acknowledge that the formulation presented is not our original proposal; therefore, the appropriate reference has been included. Our intention was only to provide context for the discussion.

On the other hand, the novelty of this work lies in placing, within that context that the reviewer refers to as “conventional neo-Darwinism,” the concept of hormesis. Our goal is not to redefine other established terms but rather to better position the concept of hormesis within this evolutionary framework. However, thanks to the reviewers’ valuable input, we have incorporated additional topics related to current mechanisms that can explain hormesis, including genetic and phenotypic mechanisms, as well as aspects of epigenetic regulation (Section 7). In addition, several references kindly provided by another reviewer were included to complement current topics related to variation, which were added at the end of that section.

We are in complete agreement that variation arises from the genetic code, environmental influences, and the interaction between both factors. In addition, a reference to Lynch (2007) was included, in which it is mentioned that natural selection acts only as a filter, as indicated by the reviewer.

In the revised manuscript (Section 7), we have expanded on the topics kindly suggested by the reviewer, such as cognition-based evolution, the microbiome, mitohormesis, dysbiosis, the pathobiome, and their roles in adaptation.

Comment 4:

A few minor criticisms where amendments should or might be considered are noted to follow:

  1. a) The reviewer is unclear why Weismann's ideas are being discussed, since they have been proven wrong, see

Noble, D. (2021). The illusions of the modern synthesis. Biosemiotics, 14(1), 5-24. If these are going to be discussed, then it is pertinent to mention that his notions are now considered invalid.

  1. b) The authors state: "It also explicitly includes that the stress factor is environmental, which broadens the conceptual framework beyond chemical or pharmacological agents." Aren't chemical and pharmacological agents in the environment? Aren't some of them actual or potential mutagens? The reviewer does not understand the logic of this statement. Furthermore, the integration of hormesis into an evolutionary framework, extending it beyond toxicology, has been going on for decades, as acknowledged by the authors in their historical review. How is this novel?
  2. c) Just as a point of discussion, the authors mention Darwin's attitude about selection and 'perfection'. They attribute this to Darwin: " The final result of natural selection tends to become more and more perfected in relation to environmental conditions." It is uncertain whether the authors are aware that this flat statement misleads the reader unless further context is provided. Darwin meant this term as a relative one, … one organism relative to another, but it was not intended to suggest actual perfection.

In fact, the notion that evolution leads to perfection is clearly misguided. If organisms are 'perfectly fit' within any given set of environmental conditions, they necessarily have sacrificed robustness. Just to note, evolution is never about perfection, nor is it actually 'survival of the fittest', a trope that has long outlived its usefulness. Evolution is always about being 'fit enough to survive, ' which better maintains the potential variations that remain 'on-call' to meet unpredictable shifts in environmental conditions.

  1. d) p. 11 " The importance of changes within the environment in which individuals develop for the production of phenotypic variability is imminent." Should that be 'eminent'. How is it imminent?
  2. e) Also, in the figures, shouldn't 'genotipe' be 'genotype'?

In sum, an interesting article, which at its base is a useful review of historical thinking about hormesis. To be suitable for publication with respect to novelty, the authors need to consider several amendments, relating to improving the clarity of their conceptualization and creating a more workable definition. One suggestion for that pathway would be to focus on cellular mechanisms. The authors might find others. As an example, in the discussion and conclusion, the authors indicate that their approach differs from others by emphasizing a definition that "delves deeper into the sequence of internal changes the organism, emphasizing the temporal and progressive nature of the response." At issue is that the manuscript, as it is written, does not demonstrate how those processes operate within a 'new' definition, and further, that definition remains problematic.  For example, this statement is made: " A local stimulus can induce beneficial changes at the systemic level, showing that hormesis is a phenomenon that transcends the local response from which it originates." Certainly, but how does this happen within this new definitional context, and how is that different, since this is the essence of plasticity?  In that way, the novelty of any new framework would be more clearly revealed to the readership, and the definitional terms would crystallize. In the reviewer's opinion, the best approach would be to include additional information that focuses on cellular mechanisms that better illustrate how hormesis relates to adaptive flexibility.

Response 4:

  1. a) The authors acknowledge the reviewer’s observation regarding Weismann’s concept. Accordingly, this content was removed in order to place greater emphasis on the description of the molecular mechanisms.
  2. b) We agree with the reviewer’s corrections; therefore, that information was removed. In any case, part of the environmental factors may include newly synthesized chemical compounds.
  3. c) The authors agree with the reviewer’s comments; therefore, that information was deleted.
  4. d) The corresponding correction was made.
  5. e) The corresponding correction was made.

Reviewer 3 Report

Comments and Suggestions for Authors

The paper provides the support to formally include the concept within the evolutionary biology ideas by showing its relatedness and coherency with the latter.  In this sense the paper is an interesting proposal that deserves publication. However, I have several comments in order to make it more consistent and concise.  Please see the specific comments below

  1. adaptation

Page 4 final second paragraph. Hormesis is described as part of canalization as this strategy responds to the environmental challenge allowing survival. But, how can you differentiate hormesis from homeostasis? Homeostasis requires challenges from the environment to express too. Please detail the differences.

Page 5 second paragraph. Authors discuss the lack of fitting for hormesis and genetic adaptation since no genetic changes are under the process, but what could be the relationship between hormesis and phenotypic plasticity as this very concept starts from the same genetic background but adaptive variation. Latter sections work on the topic and somehow contradict the tone of the current section.

2.1. Subsection Role that hormesis play in natural selection

Please consider this subtitle, is “subsection” required? Can this title be changed to “Role of hormesis in natural selection”?

Page 5 Authors propose that adaptation is part of the hormesis process but I invite authors to consider that rather hormesis is a concept related to adaptation since adaptation is a more extensive concept that included processes where the genetic background directly responds to the challenge, or that there are adaptive responses with no biphasic behavior at all.

  1. adaptability

The first paragraph of this section needs review as the current structure requires a verb or a different writing, something like “Adaptability is the capacity…..” This comment applies for all other comments of the following sections.

  1. Variation

In a similar way to the previous section, modify the first paragraph as the current version looks disconnected from the text.

First line of the second paragraph. “Since antiquity, the Greeks recognized the existence of new "varieties" within species”. I suggest to remove the word new as variation is a standing phenomenon, clearly it appears at some point but it is salient as a fact.

Authors strongly focus on Darwin´s original ideas and proposed mechanism. This suggests a final landscape where a few variants, the fittest, survive. All these variants are supported in heritable components. However, there have been extensive advances on the role of variation on adaptation. For example, Magnon & Corbara 2022 discuss the many instances where a single peak response approach is insufficient to explain adaptation. Please do not feel that the current suggestion is a demeanor but I invite authors to consider the extensive and more complex descriptions presented in reviews such as those of Orr 2005 and Lynch 2007.  The extended evolutionary synthesis proposed by Pigliucci, as authors mention in the final sections, calls for the inclusion of many elements beyond the DNA. There you can find several coincidences and concepts that may embrace hormesis too.

- Magnon, V., Corbara, B. 2022. When the “satisficing” is the new “fittest”: how a proscriptive definition of adaptation can change our view of cognition and culture. The Science of Nature 109:42. https://doi.org/10.1007/s00114-022-01814-9

- Orr, A. 2005. The genetic theory of adaptation: a brief history. Nature reviews genetics 6: 119-127.

- Lynch, M. 2007. The frailty of adaptive hypotheses for the origins of organismal complexity. PNAS 104 (1): 8597-8604.

I also suggest to change the expression of “individual variation” as variation is a property of the population, not of the individual. Variation is only understood when you have a reference, i. e., other individuals.

Page 8, first paragraph. Again, authors focus on original ideas such as Weismann´s but there are extensive advances over the role of genetic variants on the survival and evolution. The infinitesimal model consolidated by Fischer has been extensively reviewed and today, many authors call the attention to higher impact genes as the more likely to have a significant impact on adaptation. This without considering the complex interaction between genes that certainly influence the impact of each variant. See for example, De Visser & Krugg 2014 for a classic in the topic. A deeper emphasis on the role of epistasis and non-DNA mechanisms is more appropriate as authors develop after the seventh line.

I believe this section, and the others below that extend the historical context, can be significantly shortened as there are multiple developments that facilitate the inclusion of the hormesis idea without a deep justification and analysis of the conceptual framework in actual evolutionary biology. The final paragraph of this section summarizes many of the current points

6 Plasticity

Authors attribute adaptive properties to plasticity but this is a questionable approach. Phenotypic plasticity (PP) has the same characteristics as genetic variation in this regard. It can be good or it can be bad for the population or the individual. Please check that standard definitions of plasticity do not include the attribute of adaptation. The fitness impact of PP is a second step and it is highly related to the context and the processes occurring.  At the final paragraph of the section authors quote West-Eberhard whom has pioneering and leading the idea of plasticity, that author do not include the adaptive notion in the definition either.

Figures require a deeper use as in several cases these are just mentioned but no explained.

Author Response

Comment 1:

  1. Adaptation

Page 4 final second paragraph. Hormesis is described as part of canalization as this strategy responds to the environmental challenge allowing survival. But, how can you differentiate hormesis from homeostasis? Homeostasis requires challenges from the environment to express too. Please detail the differences.

Page 5 second paragraph. Authors discuss the lack of fitting for hormesis and genetic adaptation since no genetic changes are under the process, but what could be the relationship between hormesis and phenotypic plasticity as this very concept starts from the same genetic background but adaptive variation. Latter sections work on the topic and somehow contradict the tone of the current section.

2.1. Subsection Role that hormesis play in natural selection

Please consider this subtitle, is “subsection” required? Can this title be changed to “Role of hormesis in natural selection”?

Page 5 Authors propose that adaptation is part of the hormesis process but I invite authors to consider that rather hormesis is a concept related to adaptation since adaptation is a more extensive concept that included processes where the genetic background directly responds to the challenge, or that there are adaptive responses with no biphasic behavior at all.

Response 1:

  1. a) a) The manuscript details the differentiation between hormesis and homeostasis. We believe this distinction is essential, which is why it is explicitly addressed. Basically, homeostasis represents a dynamic equilibrium in which fluctuations over time are minimal. In contrast, hormesis involves larger changes that can disrupt the intrinsic dynamic rhythm of the species—that is, homeostasis itself.

Homeostasis reflects the normal state of the organism’s ontogenic condition and is closely related to time, meaning that it varies only minimally from one moment to another. Hormesis, on the other hand, alters this homeostatic balance. In some individuals, canalization may form part of the hormetic process, in which alterations—although capable of disrupting homeostasis—still allow the organism to maintain the ability to produce a stable phenotype despite genetic or environmental variation, whereas homeostasis maintains stable internal conditions.

  1. b) We thank the reviewer for the comment. In the revised manuscript, we have clarified that hormesis can operate not only at the phenotypic level but also through genetic mechanisms, as supported by recent studies. In the case of phenotypic mechanisms (e.g., physiology, metabolism, or cellular defenses), these improve fitness temporarily and can be critical for survival under fluctuating environmental conditions. Moreover, the fact that these phenotypic adaptations indirectly influence evolution by allowing the survival of organisms while mutations arise enables the genetic assimilation of the trait or shapes the environment-dependent fitness landscapes. This consideration was made taking into account some of the references kindly provided by the editor.
  2. c) The word “subsection” was removed.
  3. d) Furthermore, the authors fully agree that hormesis is a process related to adaptation. We also agree that hormesis represents one of the different ways in which adaptation can occur; therefore, the corresponding correction was made.

Comment 2:

  1. adaptability

The first paragraph of this section needs review as the current structure requires a verb or a different writing, something like “Adaptability is the capacity…..” This comment applies for all other comments of the following sections.

Response 2:

The appropriate corrections have been made.

Comment 3:

  1. Variation

In a similar way to the previous section, modify the first paragraph as the current version looks disconnected from the text.

First line of the second paragraph. “Since antiquity, the Greeks recognized the existence of new "varieties" within species”. I suggest to remove the word new as variation is a standing phenomenon, clearly it appears at some point but it is salient as a fact.

Authors strongly focus on Darwin´s original ideas and proposed mechanism. This suggests a final landscape where a few variants, the fittest, survive. All these variants are supported in heritable components. However, there have been extensive advances on the role of variation on adaptation. For example, Magnon & Corbara 2022 discuss the many instances where a single peak response approach is insufficient to explain adaptation. Please do not feel that the current suggestion is a demeanor but I invite authors to consider the extensive and more complex descriptions presented in reviews such as those of Orr 2005 and Lynch 2007.  The extended evolutionary synthesis proposed by Pigliucci, as authors mention in the final sections, calls for the inclusion of many elements beyond the DNA. There you can find several coincidences and concepts that may embrace hormesis too.

- Magnon, V., Corbara, B. 2022. When the “satisficing” is the new “fittest”: how a proscriptive definition of adaptation can change our view of cognition and culture. The Science of Nature 109:42. https://doi.org/10.1007/s00114-022-01814-9

- Orr, A. 2005. The genetic theory of adaptation: a brief history. Nature reviews genetics 6: 119-127.

- Lynch, M. 2007. The frailty of adaptive hypotheses for the origins of organismal complexity. PNAS 104 (1): 8597-8604.

I also suggest to change the expression of “individual variation” as variation is a property of the population, not of the individual. Variation is only understood when you have a reference, i. e., other individuals.

Page 8, first paragraph. Again, authors focus on original ideas such as Weismann´s but there are extensive advances over the role of genetic variants on the survival and evolution. The infinitesimal model consolidated by Fischer has been extensively reviewed and today, many authors call the attention to higher impact genes as the more likely to have a significant impact on adaptation. This without considering the complex interaction between genes that certainly influence the impact of each variant. See for example, De Visser & Krugg 2014 for a classic in the topic. A deeper emphasis on the role of epistasis and non-DNA mechanisms is more appropriate as authors develop after the seventh line.

I believe this section, and the others below that extend the historical context, can be significantly shortened as there are multiple developments that facilitate the inclusion of the hormesis idea without a deep justification and analysis of the conceptual framework in actual evolutionary biology. The final paragraph of this section summarizes many of the current points

Response 3:

  1. a) The correction was made in the first paragraph.
  2. b) The word “new” was removed.
  3. c) The authors incorporated the editor’s suggested references; however, the considerations of Magnon and Corbara were placed at the end of the section on adaptation. The contributions of Orr (2005) and Lynch (2007) were also included in the manuscript, as we believe they add significant value to the topic. In Section 7, various molecular mechanisms are detailed in which the hormetic process may occur, both at the DNA level and through mechanisms beyond genetic material.
  4. d) The reviewer’s suggestion to change “individual variation” to “variation” was implemented.
  5. e) We again acknowledge that the references provided by the reviewer are highly relevant; therefore, they were incorporated into the text of the manuscript.
  6. f) The length of the section was reduced by removing Weismann’s contributions, as requested by another reviewer. We believe that during the manuscript preparation process, we already summarized the authors’ contributions substantially.

Comment 4:

6 Plasticity

Authors attribute adaptive properties to plasticity but this is a questionable approach. Phenotypic plasticity (PP) has the same characteristics as genetic variation in this regard. It can be good or it can be bad for the population or the individual. Please check that standard definitions of plasticity do not include the attribute of adaptation. The fitness impact of PP is a second step and it is highly related to the context and the processes occurring.  At the final paragraph of the section authors quote West-Eberhard whom has pioneering and leading the idea of plasticity, that author do not include the adaptive notion in the definition either.

Figures require a deeper use as in several cases these are just mentioned but no explained.

Response 4:

The term ‘plasticity’ was adjusted according to the reviewer’s observations.

Round 2

Reviewer 2 Report

Comments and Suggestions for Authors

The reviewer thanks the authors for their careful responses to the review critique, and believe that the publication is warranted after these thoughtful amendments. The addition of the discussion was well done and particularly helpful.

Two final trivial amendments are requested.

On p. 11, the statement is made: "Although Darwin's genetic understanding of adaptation …... " Darwin had no notion of genes and this should be changed accordingly.

On page 21, in the discussion in the middle of the page, 'homeostasis' should be changed to 'homeorhetic'. This latter term emphasizes that cells are never in 'equilbrium' with their external environment, which has become the unfortunate connotation of 'homeostasis'. Instead, cells are in constant kinetic flux, i.e. dynamic kinetic stability.

Author Response

Comments

Two final trivial amendments are requested.

On p. 11, the statement is made: "Although Darwin's genetic understanding of adaptation …... " Darwin had no notion of genes and this should be changed accordingly.

On page 21, in the discussion in the middle of the page, 'homeostasis' should be changed to 'homeorhetic'. This latter term emphasizes that cells are never in 'equilbrium' with their external environment, which has become the unfortunate connotation of 'homeostasis'. Instead, cells are in constant kinetic flux, i.e. dynamic kinetic stability.

Response 1:

Page 11. The authors agree with the reviewer and modified the meaning of the statement:

“Although Darwin conceived adaptation as a gradual process driven by the natural selection of small incremental changes, the early Mendelians, such as William Bateson and Hugo de Vries, ….”

Page 21. The term homeostasis was replaced with homeorhesis. Additionally, the definition of homeorhesis was added to the text to provide greater clarity to the context of the discussion.
